# The Impact of Concurrent Chronic Heart Failure and Chronic Kidney Dysfunction on Post-Stroke Rehabilitation Outcomes

**DOI:** 10.3390/neurolint17050070

**Published:** 2025-05-03

**Authors:** Azadeh Fischer, Nadja Jauert, Martin Schikora, Michael Joebges, Wolfram Doehner

**Affiliations:** 1Deutsches Herzzentrum der Charité, Department of Cardiology Angiology and Intensive Care Medicine (Campus Virchow), Charité Universitätsmedizin Berlin, 10117 Berlin, Germany; 2German Centre for Cardiovascular Research (DZHK), Partner Site Berlin, 13092 Berlin, Germany; 3Center for Stroke Research Berlin (CSB), Charite Universitätsmedizin,13353 Berlin, Germany; 4Department of Cardiology, Brandenburg Klinik, 16321 Bernau, Germany; schikora@brandenburgklinik.de (M.S.);; 5Kliniken Schmieder, 78464 Konstanz, Germany; 6Berlin Institute of Health Center for Regenerative Therapies, Charité–Universitätsmedizin Berlin, 10117 Berlin, Germany

**Keywords:** chronic heart failure, chronic kidney dysfunction, stroke, rehabilitation, outcome

## Abstract

**Background/Objectives:** The aim of this study was to evaluate the impact of chronic heart failure (CHF), chronic kidney dysfunction (CKD), and the combined CHF-CKD comorbidity on the outcomes of rehabilitation in stroke patients. **Methods:** A total of 586 patients who had suffered a stroke (mean age, 70 ± 13; 47.6% female; 72.4% ischemic and 27.6% hemorrhagic strokes) and who were admitted immediately after acute stroke care to a rehabilitation center were included in this cohort study and followed up with until their death or discharge from the rehabilitation center. The clinical characteristics of the patients were obtained from their medical records. The relationship between the background comorbidities (CHF, CKD, and concurrent CHF-CKD) and fatal and non-fatal unfavorable outcomes (emergency readmission to a primary hospital or transfer to a long-term care facility in a vegetative or minimally conscious state) were investigated. **Results:** Unfavorable outcomes were more common in the groups with background CHF and/or CKD. From the Cox multivariate analysis, both CHF and CKD were independent prognostic factors for the occurrence of unfavorable outcomes, with a hazard ratio (HR) of 2.28 (95% CI = 1.2–4.29; *p*-value = 0.01) and 2.19 (95% CI = 1.24–3.87; *p*-value = 0.007), respectively. Moreover, the combined CHF-CKD comorbidity showed a more than 5-fold increased risk of an adverse post-stroke outcome (HR of 5.8; 95% CI = 2.5–13.44; *p*-value < 0.001). **Conclusions:** The combined CHF-CKD comorbidity is an important independent complicating factor that, along with other known influencing factors, can affect unfavorable post-stroke outcomes more than CHF or CKD alone, and necessitates critical attention to its diagnosis and management as a separate mixed syndrome.

## 1. Introduction

In the last two decades, stroke has remained the second leading cause of death and the third leading cause of disability worldwide [1]. Despite a decline in ischemic stroke incidence, death, and disability-adjusted life years lost (DALYs) from 1990 to 2013, the global number of individuals affected by stroke or remaining disabled after a stroke has increased [2,3].

Chronic heart failure (CHF) and chronic renal dysfunction (CKD) have recently garnered significant attention as potential risk factors that contribute to increased morbidity and mortality after a stroke [4,5,6,7,8,9,10]. The combination of heart failure and stroke represents a high-risk condition that warrants specific attention, given its significant impact on adverse outcomes, delayed recovery, prolonged hospitalization, and the chronic impairment of the quality of life. CHF is also a significant comorbidity in stroke patients, contributing to worse rehabilitation outcomes and increased mortality [7].

The pathophysiological association between stroke and CKD is assumed to be mediated by endothelial and platelet dysfunction, accelerated systemic atherosclerosis, and impaired cerebral autoregulation [11,12]. These conditions not only increase the risk of stroke, but they also influence the prognosis of rehabilitation outcomes after a stroke [5,6].

Patients with CKD have a higher risk of both ischemic and hemorrhagic strokes compared to the general population. Both acute and chronic renal impairment are independently associated with worse outcomes after a stroke, including increased mortality and poorer functional recovery. End-stage kidney disease, in particular, is linked to a significantly higher incidence of ischemic and hemorrhagic strokes, as well as more severe neurological outcomes and elevated mortality rates [13,14].

Furthermore, the combination of CKD and CHF is a state that dramatically worsens the outcome [4]. Despite these considerations, limited studies have specifically addressed the independent impact of a concurrent CHF-CKD status on post-acute stroke outcomes. The pathophysiological mechanisms linking CHF and CKD to stroke outcomes include inflammation, oxidative stress, neurohormonal imbalances, and the formation of uremic toxins, which can exacerbate cerebrovascular damage and impair recovery [15]. This study aims to evaluate the impact of pre-existing comorbidities, specifically CHF, CKD, and a combined CHF-CKD state, on post-acute stroke outcomes and mortality during early rehabilitation. This investigation intends to contribute further insights into the intricate relationship between these comorbidities and post-stroke rehabilitation, shedding light on potential avenues for tailored interventions and improved patient care.

## 2. Materials and Methods

### 2.1. Ethics Committee Approval

Patients were included in this study after they or their relatives provided written informed consent. This study was performed in accordance with appropriate guidelines and approved by the Ethics Committee of the Brandenburg Medical Association before implementation (No. S2(a)2015, 6.5.2015) and has been conducted in accordance with the principles set forth in the Helsinki Declaration.

### 2.2. Patient Population

In this prospective cohort study, consecutive patients over 18 years of age with a primary diagnosis of stroke who were admitted to the rehabilitation center in Brandenburgklink Bernau, Germany, were enrolled. Stroke was defined as the acute onset of focal or global neurological dysfunction lasting more than 24 h, confirmed by both clinical and imaging means. All the patients were admitted to an inpatient early rehabilitation center directly after hospital discharge. Patients with any of the following conditions were excluded from this study:Age < 18 years old.Brain mass.Cerebral injury due to trauma.Central nervous system infection.Neurological degenerative disorders.Acute kidney injury (AKI) on admission (an increase in serum Cr by ≥ 0.3 mg/dL within 48 h; or an increase in serum Cr to ≥1.5 times baseline within the prior 7 days; or urine volume ≤ 0.5 mL/kg/h for 6 h) [16].Acute decompensated heart failure (HF) on admission (new or worsening signs and symptoms of HF).

The clinical characteristics of patients were retrieved from medical records at the Brandenburg Rehabilitation Center. The Early Rehabilitation Barthel Index (ERBI) was used to assess the patients’ neurological status. ERBI values were categorized into four classes based on the German ICD-10 catalog (an ERBI range of −325 to −201 is total dependency, −200 to −76 is severe dependency, −75 to 30 is moderate dependency, and 31 to 100 is slight dependency) [17]. CHF was diagnosed by cardiologists based on symptoms of heart failure (New York Heart Association classes II to IV) [18] and treated accordingly by the attending physician in accordance with clinical evaluation and guidelines [19]. Patients with both reduced left ventricular function and preserved ventricular function were included in this study. Atrial fibrillation (AF) was diagnosed by a past medical history of AF and new-onset AF during hospitalization, as supported by a positive electrocardiogram (ECG) [20]. Laboratory results were obtained from patients’ serum samples on admission using the standard laboratory assessments of the hospital. Estimated glomerular filtration rate (eGFR) was calculated by the Modification of Diet in Renal Disease (MDRD) equation, which is eGFR = 186 × [serum creatinine]^−1.154^ × age^−0.203^ × 0.742 (if female); CKD was defined as eGFR < 60 mL/min/1.73 m^2^ for ≥3 months [21].

Patients were categorized into four groups based on the presence of CHF and/or CKD. Group 1 included patients with no CHF and no CKD (nCHF + nCKD), Group 2 comprised those with CHF and no CKD (CHF + nCKD), Group 3 patients had CKD but no CHF (nCHF + CKD), and Group 4 were those with combined CHF and CKD (CHF + CKD). The patients were followed up with until their death or discharge from the rehabilitation center. This study’s patient follow-up ranged from 0.5 to 11 months, with a median follow-up of six months.

### 2.3. Study Endpoint

This study’s endpoint was an unfavorable outcome during in-patient rehabilitation, defined as a composite outcome of all-cause mortality, emergency re-admission to a primary hospital due to clinical deterioration, or transfer to a long-term care facility in a vegetative or minimally conscious state.

### 2.4. Statistical Analysis

Statistical analyses were carried out to compare stroke patients in four groups of this study in terms of demographics, laboratory test results, and functional assessments. The proportions and numerical variables were compared using a chi-square test and a T-test, respectively. To determine the probability of event-free survival, the Kaplan–Meier product limit method was used. For Kaplan–Meier plots, patients discharged to home or nursing home were treated as alive at 180 days during follow-up, while follow-up was not extended for patients transferred to long-term care facility in a vegetative or minimally conscious state or readmitted to a primary hospital due to clinical deterioration.

To assess the factors influencing unfavorable outcomes, variables that showed significance in the univariate analyses with *p*-values less than 0.1 were considered candidates for inclusion in the multivariate Cox proportional hazards model. In the final multivariate analyses, statistical significance was determined at a threshold of *p* < 0.05. The associations are presented as hazard ratios (HRs), along with their respective 95% confidence intervals (95% CI). IBM SPSS software, version 22, was used to perform statistical analyses in this study.

## 3. Results

A total of 597 patients in the post-acute phase after a stroke were initially considered for this study. After excluding 9 patients, a total of 586 patients [307 men and 279 women; 426 with ischemic strokes and 162 with hemorrhagic strokes; mean age, 69.73 ± 12.89 years] were included in this study. The patients were followed up with until they met one of this study’s endpoints, which were either death, re-admission to a primary hospital, transfer to a long-term care facility, or discharge from the rehabilitation center (Figure 1).

During a median follow-up of six [0.5–11] months, 40 (6.8%) patients died, 29 (4.9%) were readmitted to a primary hospital because of clinical deterioration, and 3 (0.5%) were referred to a long-term care facility in a vegetative or minimally conscious state.

The results of the Cox multivariate model analysis showed that CHF (HR = 2.28; 95% CI = 1.2–4.29; *p*-value = 0.01) and CKD (HR = 2.19; 95% CI = 1.24–3.87; *p*-value = 0.007) were two independent prognostic factors for the occurrence of fatal and non-fatal unfavorable outcomes, adjusting for age, gender, serum CRP level, and ERBI on admission (Table 1).

A total of 449 (76.6%) subjects had no CHF or CKD (Group 1), while 30 (5.1%), 86 (14.7%), and 21 (3.6%) patients had isolated CHF (Group 2), isolated CKD (Group 3), and combined CHF-CKD (Group 4), respectively. These four groups were compared in terms of their demographic characteristics, underlying risk factors, and laboratory reports at the time of admission to the rehabilitation center (Table 2). The impact of the combined CHF-CKD comorbidity was assessed using a four-group model, with the patients without either comorbidity serving as the reference group. After adjusting for age, gender, serum CRP level, and ERBI on admission, Groups 3 and 4 had higher likelihoods of an unfavorable outcome compared to Group 1, with HRs of 1.96 (95% CI 1.01–3.8) and 5.8 (95% CI 2.5–13.44), respectively (Table 3). Therefore, co-existing CHF-CKD with an HR of 5.8 (95% CI = 2.5–13.44; *p*-value < 0.001) is an important independent factor that influences post-stroke outcomes during rehabilitation.

The Kaplan–Meier analysis revealed significant differences in the mean event-free survival times among the groups. The Group 1 patients exhibited a mean survival time of 9.53 months (95% CI 9.04–10.02), Group 2 had a mean survival time of 6.02 months (4.77–7.27), Group 3 showed 6.69 months (5.47–7.91), while Group 4 had the shortest mean survival time at 3.79 months (3–4.57) (*p*-value < 0.001) (Figure 2).

## 4. Discussion

Our study shows that in patients after an acute stroke, the combined CHF-CKD comorbidity emerges as a significant independent factor, exerting a more pronounced effect on post-stroke unfavorable outcomes during rehabilitation compared to isolated CHF or CKD. While both HF and CKD independently impact adverse outcomes, the combined comorbidity of CHF and CKD stands out as an independent predictor for unfavorable outcomes during the early rehabilitation phase following an acute stroke.

After adjusting for other risk factors, we observed that both CHF and CKD are associated with a two-fold increased risk of adverse outcomes following an acute stroke [CHF—HR: 2.28 (95% CI = 1.21–4.29); CKD—HR: 2.19 (95% CI = 1.24–3.87); both *p*-values < 0.01]. Notably, the co-existence of CHF-CKD demonstrates a particularly strong association with a more than five-fold increased risk of unfavorable outcomes (HR: 5.8 (95% CI = 2.5–13.44; *p*-value < 0.001)). While the global prevalence of chronic heart failure (CHF) and chronic kidney disease (CKD) is reported to be approximately 2% and 10%, respectively, in the general adult population [22,23], our study identifies their heightened prevalence among stroke patients, with rates of 8.7% for CHF and 18.3% for CKD.

Tsagalis et al. [4] assessed the long-term prognosis of combined CHF-CKD after an acute stroke over 10 years, while our study uniquely centers on evaluating the same prognostic value during the early rehabilitation phase in post-acute stroke patients. Although the CHF-CKD condition is less common in our cohort than in Tsagalis et al.’s study among stroke patients (prevalence 3.5% vs. 7.9%, respectively), the higher HR of CHF-CKD compared to isolated CHF or CKD in both studies underscores the importance of addressing and managing this combined condition as a distinct entity. This emphasizes the necessity of addressing CHF-CKD separately, as its impact extends beyond addressing each condition in isolation.

Furthermore, the Kaplan–Meier analysis not only reinforces these findings but also highlights significant differences in the event-free survival times between the groups, emphasizing the impact of chronic heart failure (CHF), chronic kidney disease (CKD), and their combination on post-stroke survival trajectories. Notably, the patients with both CHF and CKD exhibited the shortest mean survival time, providing clear evidence of the heightened risk associated with this co-existing condition.

A notable strength of our study lies in the inclusion of data on rehabilitation and post-acute care. This information was adjusted for the functional level and disability scales of the stroke patients employing the ERBI, a reliable and valid scale specifically designed for assessing the early neurological rehabilitation of patients [17].

Our study has some limitations. First, the MDRD is the most common equation used for estimating the GFR in many clinical studies; however, it is not accurate enough at higher levels of kidney function and may account for some underdiagnosing of mild renal dysfunction [24]. Also, we did not assess the severity or duration of the comorbid conditions within our study population, factors that could have potentially influenced the observed outcomes.

Despite these limitations, our study demonstrates that the CHF-CKD condition is an independent and significant predictor of unfavorable outcomes, such as mortality, re-admission to a primary hospital, or transfer to a long-term care facility in a vegetative or minimally conscious state. These research findings reveal important insights into the associations between these conditions and unfavorable outcomes, contributing to a broader understanding of stroke recovery. These results suggest that proactively addressing or effectively managing comorbid conditions, specifically congestive heart failure (CHF) and chronic kidney disease (CKD), may enhance the outcomes of rehabilitation in stroke patients. Furthermore, these results have implications for clinical practice, emphasizing the importance of tailored rehabilitation strategies for patients with co-existing CHF-CKD.

## 5. Conclusions

Co-existing CHF-CKD is an important independent factor, exerting a more pronounced impact on post-stroke unfavorable outcomes compared to isolated CHF or CKD. This underscores the importance of concentrated efforts in both diagnosing and managing this comorbid condition as a unique mixed syndrome, among other contributing factors. Furthermore, these findings enhance our understanding of the factors influencing patient outcomes, paving the way for informed decision-making in clinical settings and the development of targeted interventions for improved post-stroke care.

## Figures and Tables

**Figure 1 neurolint-17-00070-f001:**
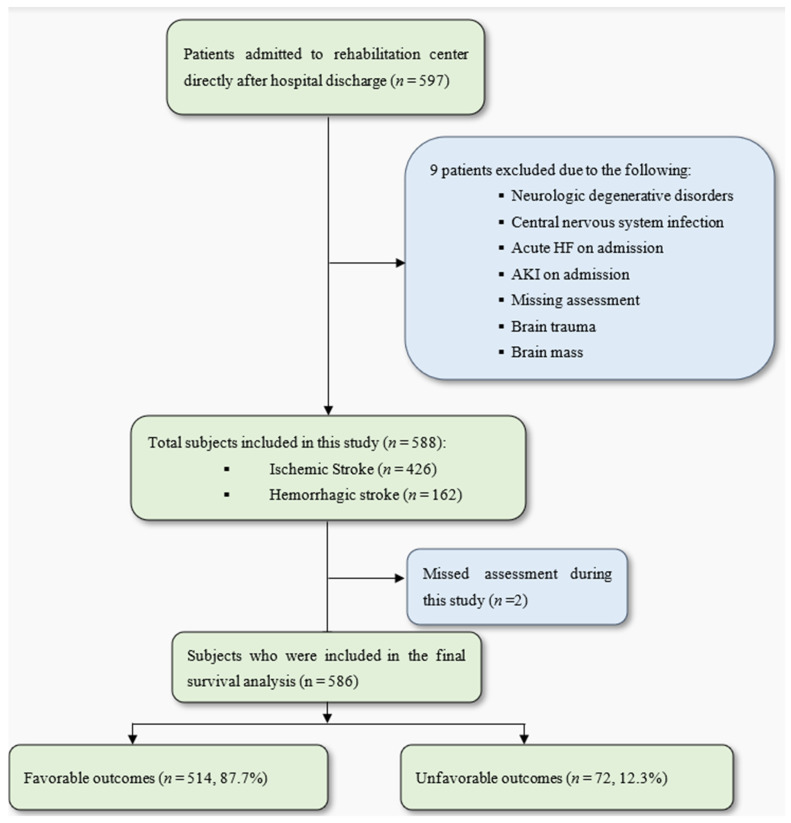
Study flowchart. AKI: acute kidney disease; HF: heart failure. A favorable outcome was considered a discharge to home or a nursing home. Unfavorable outcomes were defined as a composite outcome of all-cause mortality, re-admission to a primary hospital, or transfer to a long-term care facility in a vegetative or minimally conscious state.

**Figure 2 neurolint-17-00070-f002:**
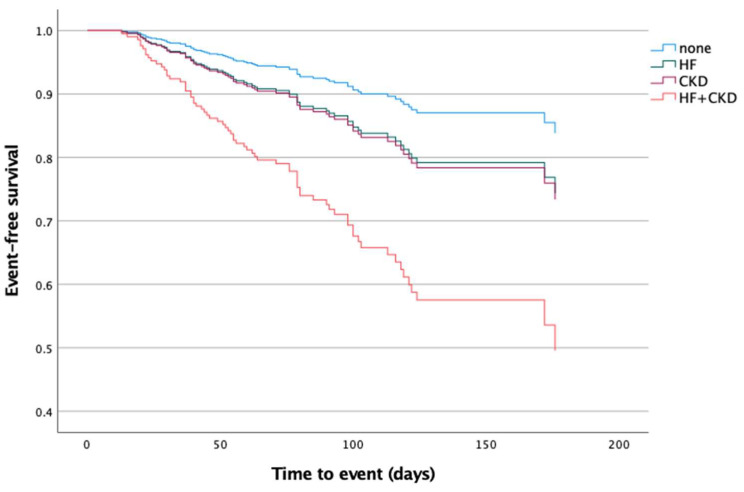
Kaplan–Meier estimates of cumulative event-free survival in post-acute stroke patients during the follow-up period based on the presence of background CHF and CKD on admission to the rehabilitation center. HF: heart failure; CKD: chronic kidney dysfunction.

**Table 1 neurolint-17-00070-t001:** Factors affecting unfavorable outcomes during follow-up period from multivariate Cox proportional hazard analysis of this study’s population.

Dependent Variable	Composite Unfavorable Outcome [HR (95% CI)] *	*p*-Value
CHF	2.28 (1.21–4.29)	0.01
CKD	2.19 (1.24–3.87)	0.007

HR: hazard ratio; CI: confidence interval; CHF: chronic heart failure; CKD: chronic kidney dysfunction. * The HR was adjusted for age, gender, serum CRP level, and ERBI on admission.

**Table 2 neurolint-17-00070-t002:** Demographic, clinical, and biochemical characteristics of patients on admission to the rehabilitation center after acute stroke based on the presence of background CHF and/or CKD.

Parameters	No CHF + No CKD (*n* = 449)	CHF + No CKD (*n* = 30)	No CHF + CKD (*n* = 86)	CHF + CKD (*n* = 21)	*p*-Value
Mean age, years (SD)	68.78 (13)	69.93 (14.69)	72.79 (11.3)	77.3 (10.02)	0.003
Gender [2] (%)	233 (51.9%)	13 (43.3%)	50 (58.1%)	11 (52.4%)	NS
Disability (ERBI)	−3.28 (60.40)	−23.33 (66.33)	−1.88 (58.28)	−20 (62.72)	NS
Stroke type					
Ischemic	315 (70.2%)	24 (80%)	66 (76.7%)	19 (90.5)	NS
Hemorrhagic	134 (29.8%)	6 (20%)	20 (23.3%)	2 (9.5%)	
Comorbidities					
HTN (%)	335 (74.9%)	22 (73.3%)	77 (89.5%)	17 (81%)	0.02
DM (%)	121 (27.1%)	13 (43.3%)	40 (46.5%)	9 (42.9%)	<0.001
CVD (%)	77 (17.2%)	22 (73.3%)	31 (36%)	19 (90.5%)	<0.001
HLP (%)	105 (23.5%)	4 (13.3%)	20 (23.3%)	8 (38.1%)	NS
AF (%)	135 (30.2%)	17 (56.7%)	36 (41.9%)	13 (61.9%)	<0.001
Biochemical characteristics					
Cr (mg/L)	74.99 (40.42)	82.76 (41.92)	173.47 (160.71)	215.66 (217.67)	<0.001
GFR (mL/min)	99.74 (41.52)	88.8 (40.32)	52.79 (30.88)	50.2 (32.47)	<0.001
Total protein (g/L)	66.92 (39.45)	5.51 (7.66)	60.94 (15.21)	60.41 (6.6)	NS
CRP (mg/L)	31.46 (58.61)	31.98 (45.32)	30.21 (44.15)	22.1 (37.23)	NS
Hb (g/dL)	8.57 (16.9)	7.46 (1.14)	7.35 (1.41)	6.81 (1.28)	NS
WBC (×106/μL)	8.62 (3.08)	8.91 (3.53)	9.12 (5.13)	8.08 (2.98)	NS

Patients admitted to in-hospital rehabilitation center after acute stroke (*n* = 586). SD: standard deviation; NS: not significant; CHF: chronic heart failure; CKD: chronic kidney dysfunction; GFR: glomerular filtration rate; ERBI: Early Rehabilitation Barthel Index; HTN: hypertension; DM: diabetes mellitus; CVD: cardiovascular disease; HLP: hyperlipidemia; AF: atrial fibrillation; Cr: creatinine; CRP: C-reactive protein; Hb: hemoglobin; WBC: white blood cells.

**Table 3 neurolint-17-00070-t003:** Factors affecting unfavorable outcomes during follow-up period from multivariate Cox proportional hazard analysis based on four groups of patients.

Dependent Variable	Composite Unfavorable Outcome [HR (95% CI)] *	*p*-Value
CHF and CKD ^¶^		
No CHF + No CKD (reference point)	1	
CHF + No CKD	1.88 (0.78–4.49)	0.15
No CHF + CKD	1.96 (1.01–3.8)	0.04
CHF + CKD	5.8 (2.5–13.44)	<0.001

HR: hazard ratio, CI: confidence interval, CHF: chronic heart failure; CKD: chronic kidney dysfunction; ERBI: Early Rehabilitation Barthel Index; CRP: C-reactive protein. * The HR was adjusted for age, gender, serum CRP level, and ERBI on admission. ^¶^ Patients were divided into four groups according to the presence of CHF and/or CKD.

## Data Availability

Data is contained within the article.

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
