# Peer review of "The Impact of Concurrent Chronic Heart Failure and Chronic Kidney Dysfunction on Post-Stroke Rehabilitation Outcomes"

_2035-8377, 2025, doi:10.3390/neurolint17050070_

Round 1
Reviewer 1 Report
Comments and Suggestions for Authors
Dear authors,
you present a clear paper considering the two important risk factors for a bad outcome after stroke during early rehabilitation. The methodology and results are well presented, however I would kindly ask you to do some corrections and clarifications.
- AKI and acute decompensated HF by admission as an exclusion factor - did you mean by admission to the rehab center or to the primary hospital?
- Was FA evaluated by ECG or Holter (24h) ECG?
- please correct some grammar issues (especially in results part) and check spelling again
- most important - please give some explanation in the discussion part for the bad outcome in the CHF-CKD group, meaning possible reasons for such a result in this group according to your opinion or/and literature (recurrent stroke, heart failure, kidney failure, infections...-immunological state in general, less possibility of rehabilitation or...?)
Thank you.
Comments on the Quality of English LanguageSome spelling mistakes, some grammar changes needed. As I am not native English speaker it would be wise to consult one.
Author Response
Comment 1: AKI and acute decompensated HF by admission as an exclusion factor - did you mean by admission to the rehab center or to the primary hospital?
Response 1: by admission to the acute rehab center
Comment 2: Was FA evaluated by ECG or Holter (24h) ECG?
Response 2: AF evaluated as documented in the medical chart of the patient as a diagnosed history of AF in the past. We did not conduct any diagnostic study in the rehab center.
Comment 3: please correct some grammar issues (especially in results part) and check spelling again
Response 3: Thanks for the comment. Grammar issues corrected.
Comment 4: most important - please give some explanation in the discussion part for the bad outcome in the CHF-CKD group, meaning possible reasons for such a result in this group according to your opinion or/and literature (recurrent stroke, heart failure, kidney failure, infections...-immunological state in general, less possibility of rehabilitation or...?)
Response 4: The unfavorable outcomes during in-patient rehabilitation, defined as a composite outcome of all-cause mortality, emergency re-admission to a primary hospital due to clinical deterioration, or transfer to a long-term care facility in a vegetative or minimally conscious state.
Reviewer 2 Report
Comments and Suggestions for Authors
- In the introduction, the research background and significance are elaborated, focusing on current hotspots and unresolved issues in post - stroke rehabilitation. This highlights the study's uniqueness and innovation.
- In the methods section, patient inclusion and exclusion criteria are detailed, with precise definitions and diagnostic standards for acute heart failure and acute kidney injury to ensure research rigor and reproducibility.
- The sample size calculation process and basis are provided, indicating whether the study has sufficient statistical power to detect the impact of CHF, CKD, and their combination on rehabilitation outcomes, thus enhancing result credibility.
- In the multivariate Cox proportional hazards model analysis, apart from adjusting for age, gender, serum CRP level, and ERBI at admission, the inclusion of other potential confounding factors such as patient history and lifestyle factors is considered for a more comprehensive assessment of the independent effects of CHF and CKD on rehabilitation outcomes.
- Subgroup analysis is conducted for different stroke types (ischemic and hemorrhagic), examining whether the impacts of CHF and CKD differ.
- Besides survival curves, tables or other graphics are used to more intuitively display survival time and event rate differences among groups.
- The potential biological mechanisms of how CHF and CKD affect post - stroke rehabilitation outcomes are explored in - depth from perspectives like inflammation, oxidative stress, and vascular endothelial dysfunction, offering a theoretical basis for future research.
- Please provide the ethics approval number.
please improve your language.
Author Response
Thanks a lot for all the positive comments on our manuscript.
Comment 4: In the multivariate Cox proportional hazards model analysis, apart from adjusting for age, gender, serum CRP level, and ERBI at admission, the inclusion of other potential confounding factors such as patient history and lifestyle factors is considered for a more comprehensive.
Reply 4: Additional factors were considered for inclusion in the multivariate analysis; however, lifestyle variables were not collected in this study, and data on patient comorbidities were only partially available.
Comment 8: Please provide the ethics approval number.
Reply 8: The study has been approved by local ethic committee of Brandenburg. The letter doesn't have any specific number but we contacted them to obtain the approval number and we'll add it in final revision.
Reviewer 3 Report
Comments and Suggestions for Authors
Stroke is the primary cause of neurological impairment and death, impacting the social and economic landscape of numerous nations worldwide. A unique position is held by strokes that occur in the context of chronic heart and/or renal failure, mainly because these conditions alone or in combination can result in acute cardiovascular catastrophes, such as stroke. However, this puts these stroke patients at risk for significant mortality and disability. Creatinine levels at the time of discharge and the left ejection fraction in chronic heart failure are recognized to be indicators of a poor prognosis for patients. The choice of treatment strategies for individuals with chronic heart failure and chronic renal failure is complicated, and this issue is made worse by the presence of stroke. Therefore, the investigation of mortality predictors in individuals with a combination of stroke and chronic heart and renal disease makes the authors' work pertinent. The authors employ appropriate statistical techniques to validate their hypothesis, and the article is written in a conventional style. They also continuously support the significance of the selected research issue and the methods used to accomplish the study's objective. Only seven publications were published between 2020 and 2025, and the writers mostly used articles published more than five years ago to support the research topic and analyze the study's findings. The utilized works are still relevant, nonetheless. It is feasible to replicate such a study with a large number of patients in specialized treatment facilities and access to medical data. The use of illustrative material enables us to explain the research findings in a clear and simple manner. The writers examine the findings during the article's presentation. The authors logically conclude that the existence of kidney and heart pathology, either independently or in combination, is a poor predictor of the disease's fate in stroke patients based on the statistical analysis of the study results. The study was conducted with the Ethics Committee's consent and in accordance with the Helsinki Declaration.
The work's drawbacks include:
The abbreviation AF should be decoded in paragraph 2.2 in the Materials and Methods section, as this is where it is first referenced.
Since the HR transcription has already been provided, the author does not need to repeat it in the third paragraph on page 4.
The statistical significance of the differences between the groups that have been identified must be clarified in Table 2. Furthermore, even after accounting for the average error, the groups overlap, raising doubts about whether there is a statistically significant age difference between them.
Table 2: Creatinine: Based on the range of values in the CHF+CKD group, the data should be displayed as a median and interquartile value (Me; Q1-Q3) rather than as M±SD. It's unclear which groupings are different from one another.
Because the final paragraph is truncated in Figure 1, the font size may be reduced to explain why patients were not included in the study.
Align the signatures' font in Figure 2.
Author Response
Comment 1: The abbreviation AF should be decoded in paragraph 2.2 in the Materials and Methods section, as this is where it is first referenced.
Response 1: Thank you for pointing it out. Atrial Fibrillation (AF) added to the first abbreviation in the text.
Comment 2: Since the HR transcription has already been provided, the author does not need to repeat it in the third paragraph on page 4.
Response 2: Thank you for pointing it out. The repeat has been changed.
Comment 3: The statistical significance of the differences between the groups that have been identified must be clarified in Table 2. Furthermore, even after accounting for the average error, the groups overlap, raising doubts about whether there is a statistically significant age difference between them.
Response 3: Thanks for your comment. The age is significantly different between CHF+CKD and No CHF+CKD and CHF+ No CKD. That's why we adjusted all further analysis (Cox regression) based on age as well as other parameters like ERBI.
Comment 4: Table 2: Creatinine: Based on the range of values in the CHF+CKD group, the data should be displayed as a median and interquartile value (Me; Q1-Q3) rather than as M±SD. It's unclear which groupings are different from one another.
Response 4: Thanks for your comment. Based on KS test in SPSS, The Cr variable was parametric. That's why mean and SD was reported.
Comment 5: Because the final paragraph is truncated in Figure 1, the font size may be reduced to explain why patients were not included in the study.
Align the signatures' font in Figure 2.
Response 5: Which final paragraph are you referring to?
